# Can Presepsin Be Valuable in Reducing Unnecessary Antibiotic Exposure after Birth?

**DOI:** 10.3390/antibiotics12040695

**Published:** 2023-04-02

**Authors:** Thomas H. Dierikx, Henriëtte van Laerhoven, Sophie R. D. van der Schoor, Charlotte M. Nusman, Claire A. M. Lutterman, Roos J. S. Vliegenthart, Tim G. J. de Meij, Marc A. Benninga, Wes Onland, Anton H. van Kaam, Douwe H. Visser

**Affiliations:** 1Department of Neonatology, Amsterdam UMC Location University of Amsterdam, Meibergdreef 9, 1105 AZ Amsterdam, The Netherlands; 2Amsterdam Reproduction & Development, 1105 AZ Amsterdam, The Netherlands; 3Department of Pediatric Gastroenterology, Amsterdam UMC Location University of Amsterdam, Meibergdreef 9, 1105 AZ Amsterdam, The Netherlands; 4Amsterdam Gastroenterology Endocrinology Metabolism, 1105 AZ Amsterdam, The Netherlands; 5Department of Pediatrics, OLVG, 1105 AZ Amsterdam, The Netherlands; 6Department of Pediatrics, Flevoziekenhuis, 1315 RA Almere, The Netherlands

**Keywords:** early-onset neonatal sepsis, presepsin, diagnosis, antibiotics

## Abstract

Background: Due to a lack of rapid, accurate diagnostic tools for early-onset neonatal sepsis (EOS) at the initial suspicion, infants are often unnecessarily given antibiotics directly after birth. We aimed to determine the diagnostic accuracy of presepsin for EOS before antibiotic initiation and to investigate whether presepsin can be used to guide clinicians’ decisions on whether to start antibiotics. Methods: In this multicenter prospective observational cohort study, all infants who started on antibiotics for EOS suspicion were consecutively included. Presepsin concentrations were determined in blood samples collected at the initial EOS suspicion (t = 0). In addition to this, samples were collected at 3, 6, 12 and 24 h after the initial EOS suspicion and from the umbilical cord directly after birth. The diagnostic accuracy of presepsin was calculated. Results: A total of 333 infants were included, of whom 169 were born preterm. We included 65 term and 15 preterm EOS cases. At the initial EOS suspicion, the area under the curve (AUC) was 0.60 (95% confidence interval (CI) 0.50–0.70) in the term-born infants compared to 0.84 (95% CI 0.73–0.95) in the preterm infants. A cut-off value of 645 pg/mL resulted in a sensitivity of 100% and a specificity of 54% in the preterm infants. The presepsin concentrations in cord blood and at other time points did not differ significantly from the concentrations at the initial EOS suspicion. Conclusions: Presepsin is a biomarker with an acceptable diagnostic accuracy for EOS (culture-proven and clinical EOS) in preterm infants and might be of value in reducing antibiotic exposure after birth when appended to current EOS guidelines. However, the small number of EOS cases prevents us from drawing firm conclusions. Further research should be performed to evaluate whether appending a presepsin-guided step to current EOS guidelines leads to a safe decrease in antibiotic overtreatment and antibiotic-related morbidity.

## 1. Introduction

Sepsis is one of the leading causes of neonatal morbidity and mortality [1]. An accurate and rapid diagnosis of early-onset neonatal sepsis (EOS), defined as sepsis onset within 72 h of life, remains problematic mainly due to the non-specific signs and symptoms and the lack of reliable, timely diagnostic tools. In the Netherlands, the national EOS guideline is used for the decision to start empirical antibiotics after birth. This guideline is comparable to the NICE guideline and follows a risk-based approach, including maternal and neonatal risk factors, with a low threshold for the start of empirical antibiotic treatment [2]. Consequently, the number of newborns who receive antibiotic therapy for suspected EOS is up to 58 times higher than the number of newborns with a positive blood culture [3] Once started, antibiotic treatment is continued in about 30% of newborns despite a negative blood culture [4,5].

This unnecessary antibiotic exposure increases antibiotic resistance; leads to aberrations in microbial colonization; and increases the risks of necrotizing enterocolitis in preterm infants and long-term complications, such as asthma and obesity [6,7]. To diminish these complications, a strategy to safely reduce unnecessary antibiotic exposure in uninfected infants is urgently needed. Adding an early and accurate biomarker to the existing EOS guideline could be such a strategy. The diagnostic value of the biomarkers used in daily care, such as C-reactive protein, procalcitonin and different interleukins, has been studied for this purpose, but all lack sufficient accuracy at the initial EOS suspicion [8]. In contrast, the biomarker presepsin (a soluble CD14 subtype) seems to be promising for this purpose, as its concentrations increase rapidly after infection onset [9,10].

After the binding of bacterial ligands to the cell surface of monocytes and macrophages, CD14 sheds from the cell surface and is subject to proteolysis [11,12]. This leads to the release of various fragments and, finally, the generation of a small soluble peptide structure (64 amino acids, 13 kDa) named a soluble CD14 subtype (sCD14-ST) or presepsin [13]. The reference ranges of presepsin in healthy infants have been determined, with conflicting results on the possible differences between term- and preterm-born infants and the influence of clinical characteristics and the method of delivery [14,15,16]. Previous diagnostic studies on the diagnostic accuracy of presepsin for EOS in newborns have methodological flaws, and a clear cut-off value with a high negative predicting value is consequently still lacking [17,18].

Therefore, the primary aim of this multicenter prospective observational cohort study was to assess the diagnostic accuracy of presepsin directly after birth in all infants suspected of having EOS and to investigate whether presepsin can be used to guide clinicians’ decisions on whether to start antibiotics. The secondary aim was to evaluate presepsin concentrations over time, as concentrations in EOS cases might change [10].

## 2. Results

### 2.1. Participant Inclusions

A total of 398 participants were eligible for inclusion, of whom the parents of 65 infants did not consent to participation. A flowchart of patient inclusion is displayed in Appendix A. The baseline characteristics of the 333 included infants are given in Table 1 and Table 2 for term- and preterm-born infants, respectively. The median time from birth to collection of the first postnatal sample (t = 0) was 2.0 h (interquartile range (IQR) 1.1–5.5).

A total of 65 term-born infants and 15 preterm-born infants were classified as EOS cases. In all infants, a blood culture was collected. Three EOS cases were culture-proven cases (0.9%). All three isolated bacterial pathogens were *Streptococcus agalactiae*. The CRP concentrations during the first 48 h after the initial EOS suspicion were higher in both the term- and preterm-born EOS cases (median: 45.1 mg/L (IQR: 33.2–64.6) and 65 mg/L (IQR: 44.9–81.8), respectively) than in the controls (median 5.6 mg/L (IQR: 2.1–17.0) and 1.1 mg/L (IQR: 0.6–4.0), respectively).

### 2.2. Diagnostic Accuracy of Presepsin in Term-Born Infants

The presepsin concentrations were significantly higher in the EOS cases than in the controls directly after birth in the term-born infants (Figure 1; *p* = 0.04). The ROC curve at this time point is presented in Figure 2, and the AUC was 0.60 (95%CI [0.50–0.70]). Youden’s index was the highest at a cut-off of 874 pg/mL, with a 46% (95%CI [0.32–0.61]) sensitivity, a 74% (95%CI [63–83]) specificity, a PPV of 53% (95%CI [38–69]) and an NPV of 68% (95%CI [57–78]). At a cut-off of 307 pg/mL, the sensitivity was 100% (95%CI [93–100]), but the specificity decreased to 2% (95%CI [0–7]).

The cord blood concentrations did not differ from the concentrations in the first postnatal sample in the EOS cases (*p* = 0.77) and the controls (*p* = 0.11). A mixed-model analysis demonstrated no significant changes in the presepsin concentrations over time in both the EOS cases and the controls (*p* = 0.14 and *p* = 0.46, respectively; Figure 1A). An overview of the presepsin concentrations at all time points is given in Appendix A. The AUC and other diagnostic accuracy measures of the cord blood samples at all time points are demonstrated in Appendix A.

In the secondary center, 58 term EOS cases were recruited compared to 7 in the tertiary center. At the initial EOS suspicion, the AUC of the term-born participants recruited in the secondary center was 0.56 (95%CI [0.44–0.67]), and it was 0.82 (95% CI [0.65–0.99]) in the tertiary center.

### 2.3. Diagnostic Accuracy of Presepsin in Preterm-Born Infants

The presepsin concentrations were significantly higher in the EOS cases than in the controls directly after birth (t = 0) in the preterm infants (Figure 1; *p* < 0.001). The ROC curve at the initial EOS suspicion is presented in Figure 2 (AUC: 0.84; 95%CI [0.73–0.95]). Youden’s index was the highest at a cut-off of 855 pg/mL. The sensitivity was 87% (95%CI [60–98]), and the specificity was 68% (95%CI [58–77]), with a PPV and an NPV of 28% (95%CI [16–43]) and 97% (95%CI [90–100]), respectively. A sensitivity of 100% (95%CI [78–100]) was reached with a specificity of 54% (95%CI [44–64]) at a cut-off value of 645 pg/mL.

In the preterm-born infants, no differences were found between the cord blood concentrations and the concentrations in the first postnatal samples in the EOS cases and the controls (*p* = 0.12 and *p* = 0.14, respectively). No significant changes in the presepsin concentrations were found over time in the EOS cases (*p* = 0.92) or in the controls (*p* = 0.67) (Figure 1B). An overview of the presepsin concentrations in the preterm-born infants at all time points is given in Appendix A. The AUC and other diagnostic accuracy measures in the cord blood samples at all time points are shown in Appendix A.

A total of seven preterm EOS cases were recruited in the secondary center, and the other eight were recruited in the tertiary center. Directly after birth, the AUC values were 0.75 (95%CI [0.54–0.95]) in the secondary care center and 0.92 (95% CI 0.83–1.00) in the tertiary care center in the preterm infants. In the preterm infants with GA < 32 weeks, the AUC values were 0.98 (95%CI [0.94–1.00]) and 0.73 (95%CI [0.56–0.89]) in the preterm infants with GA between 32 and 37 weeks (Appendix A).

## 3. Discussion

In this prospective cohort study, we evaluated the diagnostic accuracy of presepsin for culture-proven and culture-negative EOS in a cohort of infants with an indication for empirical antibiotics based on the Dutch EOS guideline. This is the first study recruiting all infants who started with antibiotics for the suspicion of EOS. The results of this study show that presepsin is a biomarker with an acceptable diagnostic accuracy for EOS in preterm infants and that it can be of value in reducing antibiotic exposure after birth when appended to the Dutch EOS guideline. In the term-born infants, the diagnostic accuracy was low.

The majority of studies on presepsin in neonatal sepsis included both EOS and LOS cases [10,17,18]. However, differences in presepsin concentrations between EOS and LOS cases and differences in reference ranges with increasing postnatal age underline the importance of studying them as separate entities [9,14]. To the best of our knowledge, only six previous studies reported diagnostic accuracy measures of presepsin specifically for EOS [13,19,20,21,22,23]. Five of these studies included only term-born infants [20,22] or mainly term-born infants [19,21,23]. Three of these studies compared presepsin concentrations between clinical EOS cases and healthy controls without a suspicion of infection and reported AUC values of 0.77, 0.95 and 0.97, respectively [20,21,22]. Popsilova et al. (2023) included all infants with clinical signs of infection, but they excluded infants with possible EOS from their analysis. Presepsin could discriminate infants with probable EOS from infants with unlikely EOS, with an AUC of 0.85 [23]. Chen et al. (2017) found an AUC of 0.96 in a cohort of infants with clinical EOS and infants without a suspicion of infection. Only one study investigated the diagnostic accuracy of presepsin in preterm-born infants [13]. They included all infants undergoing a sepsis evaluation but excluded infants with potential EOS from their analysis. They observed an AUC of 0.75 at the initial EOS suspicion and an AUC of 0.92 after 12 h, in contrast to our observations.

Previous studies mainly reported diagnostic accuracy measures higher than those in our cohort. However, none of these previous studies included all patients with a suspicion of EOS consecutively. These studies were either case–control studies, comparing culture-proven EOS cases to healthy controls without a suspicion of EOS, or excluded patients with possible and/or culture-negative EOS from their analysis. Both approaches lead to bias and an overestimation of AUC. The use of these approaches allows for the inclusion of a population different from the population that this biomarker is intended to be used for in clinical practice, namely, all infants with an EOS suspicion. Consequently, these flaws limit the possibility to generalize the applicability of previous results to clinical practice [24,25,26].

A peripheral blood culture is still used as a gold standard for diagnosing EOS, but its diagnostic accuracy has been questioned since cultures obtained from infants with a clinical illness or increased inflammatory markers often remain sterile. Whether prolonged antibiotic therapy is indicated in these infants is still the subject of discussion. Due to the lack of accurate diagnostic tools for EOS and the absence of a consensus definition for clinical EOS, clinicians often (up to 30%) decide to continue antibiotic treatment despite a negative blood culture [4,5]. Presepsin may be a rapid and early biomarker for EOS. Presepsin concentrations can be measured quickly as a point-of-care test (POCT); a limited blood volume is needed, and concentrations increase early and rapidly in cases of a bacterial infection. In our cohort, almost 25% of all infants received prolonged antibiotic therapy, underlining the urgency of an internationally accepted consensus definition in order to prevent unnecessary antibiotic exposure [27]. Before implementing a biomarker in clinical care, it is pivotal to study the diagnostic accuracy in a population representative of that in clinical practice, including in both culture-positive and culture-negative EOS. In contrast to previous studies, we consecutively included all infants with a suspicion of EOS, defined cases as both culture-negative or culture-proven EOS and compared the results with infants in whom EOS was ruled out. In the preterm infants, the diagnostic accuracy of presepsin remained acceptable. This implies that, if presepsin is measured before initiating antibiotics, a high sensitivity could be achieved when using a relatively low cut-off value of 645 pg/mL, and, thus, no culture-proven or clinical EOS cases would be missed. At the same time, the specificity will still be reasonable, and antibiotics could thus be withheld in a large number of uninfected infants with an EOS suspicion, who would have started empirically on antibiotics with the current guidelines. Before appending a presepsin-guided step to the current guidelines, further research should be performed to evaluate whether implementation would indeed lead to a safe decrease in antibiotic prescriptions in preterm infants shortly after birth.

Our study shows conflicting results regarding the diagnostic accuracy of presepsin in term versus preterm infants. This difference is not completely elucidated, as this was not found in a recently performed meta-analysis [10]. One explanation is classification bias in the term infants, as we found a higher percentage of EOS cases than expected in the term infants (65/164; 39%) and a higher percentage than in the preterm infants (15/169; 9%). Due to a lack of a consensus definition for EOS, one could hypothesize that some of the uninfected term-born control infants were misclassified as EOS cases. This might be a consequence of a difference in rationale for antibiotic initiation, as preterm infants are more often started on antibiotics based solely on risk factors in the absence of a strong clinical suspicion of EOS, and this may have led to the underestimation of AUC in the term-born infants. Future studies including all infants suspected of having EOS, with predefined definitions for culture-negative EOS, are warranted to determine whether a classification bias affected our results in the term-born infants or whether presepsin is an accurate biomarker for culture-negative EOS in term-born infants.

Since the collection of blood directly after birth can be challenging, especially in low-birthweight infants [28,29], we evaluated the correlation of presepsin concentrations in umbilical cord blood and neonatal plasma samples obtained from a peripheral vein within two hours of birth. The presepsin concentrations in the umbilical cord blood in our cohort were comparable to the concentrations in the neonatal samples, as previously reported [14]. In line with our findings, a previous study reported that the discriminative ability of umbilical cord blood presepsin is high, as the presepsin concentration was higher in the cord blood of all 76 preterm EOS cases (range 1442–3988 pg/mL) than in the 212 preterm controls (range 116–326 pg/mL) in that study [30]. Therefore, non-invasively collected umbilical cord blood might be used for presepsin measurements if there is a prenatal EOS suspicion.

Presepsin concentrations may be affected by factors other than EOS, such as the route of delivery and the presence of respiratory distress syndrome (RDS). The main goal of a new EOS biomarker is to discriminate between EOS cases requiring antibiotics and uninfected controls in the population of all infants with an EOS suspicion. Although it is most important not to miss any EOS cases, it is still important to have a high specificity so that antibiotics can be withheld in uninfected cases simultaneously. We demonstrated that this is possible using a relatively low cut-off value, and factors such as RDS do not significantly impact the discriminative ability of presepsin.

The strengths of this study include the large sample size, making it possible to perform analyses stratified by gestational age. Furthermore, all infants who started antibiotics for suspected EOS were recruited consecutively and were included in the analysis, so bias was minimized, and our results provide a realistic view of the potential of this biomarker in clinical practice [25]. The longitudinal collection of samples, including umbilical cord blood, provided valuable information on the course of presepsin during the first 24 h in infants who are infected and uninfected.

The limitations of this study and other studies on biomarkers for neonatal sepsis include the lack of a consensus case definition for EOS, increasing the risk of classification bias. Furthermore, we did not compare the diagnostic accuracy of presepsin with that of other biomarkers, such as CRP, PCT and IL-6, but other studies and meta-analyses demonstrated that presepsin had a higher accuracy than these other biomarkers at the initial EOS suspicion [31,32]. The small number of EOS cases (culture-proven and clinical EOS cases) is another limitation leading to the sensitivity and specificity having wide confidence intervals.

In conclusion, presepsin is a biomarker with an acceptable diagnostic accuracy for EOS (culture-proven and clinical EOS) in preterm infants and might be of value in reducing antibiotic exposure after birth when appended to current EOS guidelines. However, the small number of EOS cases prevents us from drawing firm conclusions. Presepsin can be measured in umbilical cord blood with results comparable to those of samples taken directly after birth. Further research should be performed to evaluate whether appending a presepsin-guided step to the current EOS guidelines leads to a safe decrease in antibiotic overtreatment and antibiotic-related morbidity.

## 4. Materials and Methods

### 4.1. Participants

In this multicenter prospective observational cohort study, all infants who started with antibiotics within the first 72 h based on the Dutch EOS guideline were eligible for participation [2]. In the Dutch EOS guideline, the maternal and neonatal risk factors for EOS are categorized as red flags or minor criteria (Appendix A). In the presence of 1 red flag or ≥2 minor criteria, it is advised to draw a peripheral blood culture and initiate antibiotics for an EOS suspicion [2,33]. Infants were included if both parents gave written informed consent. Infants were not eligible in cases of a confirmed congenital infection (toxoplasmosis, rubella, cytomegalovirus infection, syphilis and herpes). The participants were consecutively recruited in one level III center (Emma Children’s Hospital) and in one level II center with two locations (OLVG East and West) between August 2018 and June 2021. The study protocol was approved by the medical ethical committee (WO 18.020).

Antibiotic treatment was discontinued after 36 h in cases of a negative blood culture, indicating a clinical condition with no clinical indicators of possible infection. Infants who received antibiotics for ≥5 days and had blood cultures with the growth of potentially pathogenic micro-organisms were classified as culture-proven EOS. For infants who continued antibiotics for ≥5 days for suspected EOS based on the clinician’s judgement and who had CRP levels ≥10 mg/L but negative blood cultures, the results were classified as clinical EOS. All other participants not meeting the criteria for culture-proven or clinical EOS were considered uninfected controls. The treatment and classification of the participants as EOS cases or as controls were carried out blinded from the presepsin measurements.

### 4.2. Study Samples

Combined with blood collection for standard care, 0.2 mL of blood was obtained before the initiation of antibiotics directly after birth at the initial EOS suspicion (t = 0) and 3, 6, 12 and 24 h afterwards in ethylenediaminetetraacetic acid (EDTA) tubes. If it was prenatally known that the infant would start on empirical antibiotics and be eligible for participation, a blood sample of the umbilical cord was also collected. The blood was centrifuged at 2000× *g* for 10 min at 18 °C. Plasma was extracted and stored at −80 °C until further handling.

After the completion of participant recruitment, the samples were thawed, and presepsin levels were measured blinded using a rapid chemiluminescent enzyme immunoassay on a PATHFAST immunoanalyzer (Mitsubishi Chemical Medience Corporation, Tokyo, Japan), a chemiluminescent immunoassay analyzer (CLEIA), according to the manufacturer’s protocol using 100 µL plasma. If <100 µL plasma was available, the samples were diluted with sodium chloride.

### 4.3. Statistical Analysis

The baseline characteristics are presented descriptively. As reference ranges differ between term and preterm infants [10,14,15,16], analyses were performed for the term- and preterm-born infants separately. The presepsin concentrations directly after birth at the initial EOS suspicion (t = 0) were compared between the EOS cases and the uninfected controls using the Mann–Whitney U test. The receiver-operating characteristic (ROC) curve was analyzed, and the area under the curve (AUC) was calculated. Youden’s index was determined, and the sensitivity, specificity, positive predictive value (PPV) and negative predictive value (NPV) were determined at this cut-off. Furthermore, the cut-off point with a maximum sensitivity of 100% and the highest possible specificity was calculated in order to determine a cut-off value at which no infected EOS cases would be missed. The 95% confidence intervals (CIs) were calculated around the diagnostic accuracy measures. Subgroup analyses were performed for the preterm-born infants with a gestational age < 32 weeks, for infants with a gestational age between 32 and 37 weeks and for the two different recruiting sites.

To evaluate whether the presepsin concentration in the umbilical cord differed from the concentration in the first neonatal sample collected postpartum, the Wilcoxon signed-rank test was used. A mixed-model analysis was performed to evaluate whether the presepsin concentrations changed during the first 24 h after antibiotic initiation. Two-tailed *p*-values of <0.05 were considered statistically significant. Statistical analyses were performed using IBM Statistical Product and Service Solutions (SPSS) for Windows Version 28 (IBM Corp., Armonk, NY, USA) and R version 4.0.3.

## Figures and Tables

**Figure 1 antibiotics-12-00695-f001:**
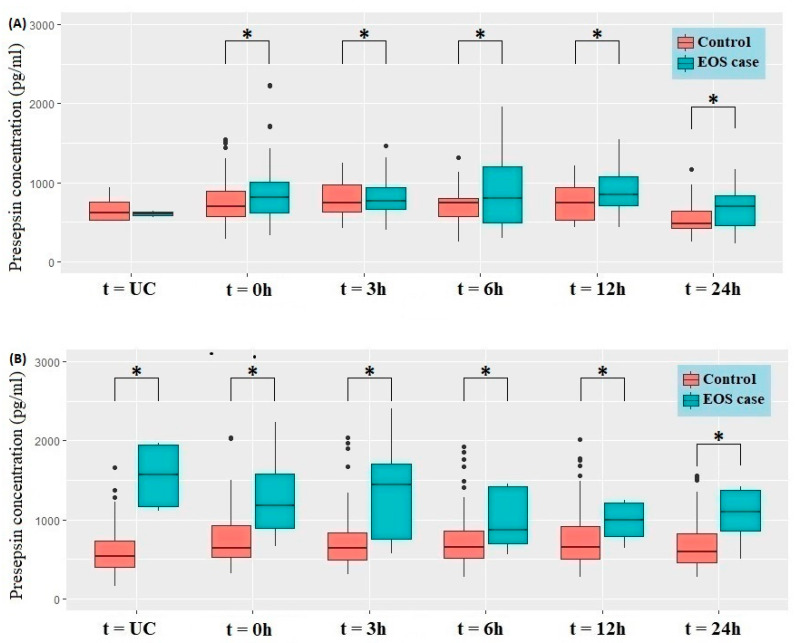
Boxplots of presepsin concentrations (pg/mL) before antibiotic initiation at initial sepsis suspicion (t = 0) and the other time points for early-onset neonatal sepsis cases (blue) and uninfected controls (red) in term-born infants (**A**) and preterm-born infants (**B**). EOS = early-onset sepsis; h = hour; UC = umbilical cord blood. Asterisks indicate a statistically significant difference in presepsin concentration between controls and EOS cases.

**Figure 2 antibiotics-12-00695-f002:**
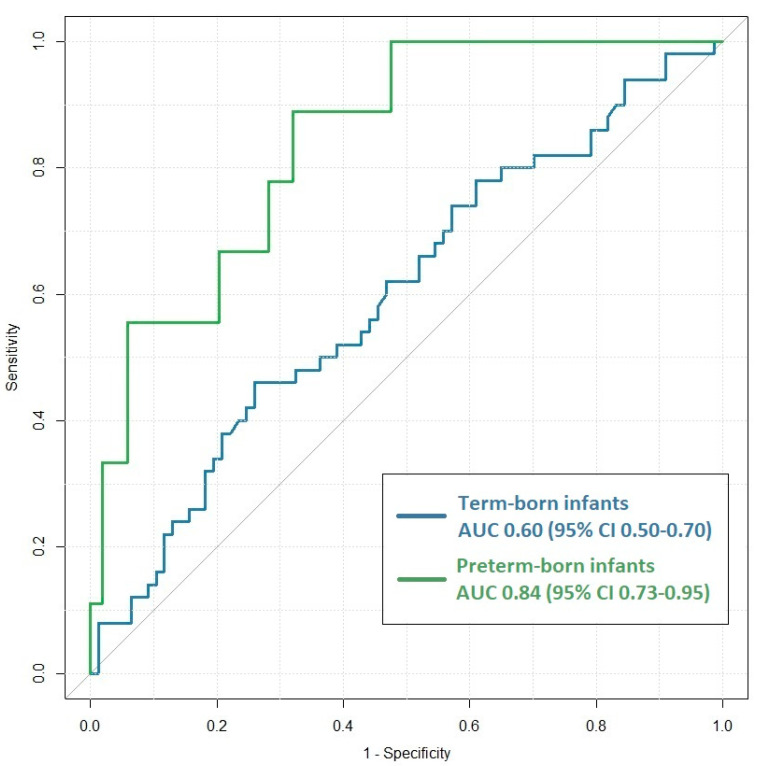
Receiver-operating characteristic (ROC) curves for presepsin concentrations before antibiotic initiation at initial sepsis suspicion (t = 0) differentiating between early-onset neonatal sepsis cases and uninfected controls in term-born infants (blue) and preterm-born infants (green). Area under the curve (AUC) values were 0.60 (95% CI: 0.50–0.70) and 0.84 (95% CI: 0.73–0.95), respectively.

**Table 1 antibiotics-12-00695-t001:** Baseline characteristics of term-born infants.

Clinical Values	Control (*n* = 99)	Case (*n* = 65)
Gestational age, median [IQR], weeks + days	40^+1^ [38^+6^–41^+3^]	40^+5^ [38^+6^–41^+3^]
Birthweight, median [IQR], grams	3534 [3208–3810]	3538 [3258–3873]
Female sex, *n* (%)	34 (34)	30 (46)
Vaginal delivery, *n* (%)	71 (72)	46 (71)
Maternal age, mean (SD), years	33.0 (4.2)	33.2 (4.9)
Admission in level III center, *n* (%)	20 (20)	7 (11)
Mother with sepsis (red flag), *n* (%)	7 (7)	4 (6)
Twin with infection (red flag), *n* (%)	0 (0)	0 (0)
Invasive GBS previous child, *n* (%)	0 (0)	1 (2)
Maternal GBS, *n* (%)	12 (12)	1 (2)
PROM ^a^, *n* (%)	40 (40)	19 (29)
Maternal fever > 38 °C, *n* (%)	30 (30)	24 (37)
Maternal intrapartum antibiotics, *n* (%)	62 (63)	40 (62)
Neonatal red flag clinical symptom, *n* (%)	15 (15)	16 (25)
Well appearing, *n* (%)	28 (28)	14 (21)

^a^ PROM defined as rupture of membranes >24 h before labor onset after a pregnancy of ≥37 weeks. IQR: interquartile range; PROM: premature rupture of membranes; SD: standard deviation.

**Table 2 antibiotics-12-00695-t002:** Baseline characteristics of preterm-born infants.

Clinical Values	Control (*n* = 154)	Case (*n* = 15)
Gestational age, median [IQR], weeks + days	31^+3^ [28^+1^–35^+3^]	45^+3^ [26^+6^–36^+3^]
Gestational age 32^+0^ to 36^+6^ weeks, *n* (%)	73 (47)	8 (53)
Gestational age < 32^+0^ weeks, *n* (%)	81 (53)	7 (47)
Birthweight, median [IQR], grams	1684 [1177–2438]	2482 [843–2826]
Female sex, *n* (%)	77 (50)	5 (33)
Vaginal delivery, *n* (%)	110 (71)	8 (53)
Maternal age, mean (SD), years	32.5 (5.2)	32.5 (3.6)
Admission in level III center, *n* (%)	92 (60)	8 (53)
Mother with sepsis(red flag), *n* (%)	2 (1)	4 (27)
Twin with infection (red flag), *n* (%)	1 (1)	0 (0)
Invasive GBS previous child, *n* (%)	0 (0)	0 (0)
Maternal GBS, *n* (%)	17 (11)	3 (20)
PPROM ^a^, *n* (%)	53 (34)	9 (60)
Spontaneous premature birth, *n* (%)	106 (69)	10 (67)
Maternal fever > 38 °C, *n* (%)	10 (7)	5 (33)
Maternal intrapartum antibiotics, *n* (%)	54 (35)	7 (47)
Neonatal red flag clinical symptom, *n* (%)	4 (3)	2 (13)
Well appearing, *n* (%)	42 (27)	1 (7)

^a^ PPROM defined as rupture of membranes >18 h before labor onset after a pregnancy of <37 weeks. IQR: interquartile range; PPROM: preterm premature rupture of membranes SD: standard deviation.

## Data Availability

De-identified data for this study are available from the corresponding author upon reasonable request.

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
