# Peer review of "Can Presepsin Be Valuable in Reducing Unnecessary Antibiotic Exposure after Birth?"

_antibiotics, 2023, doi:10.3390/antibiotics12040695_

Round 1
Reviewer 1 Report
Dear Authors,
I have read this manuscript with interest. In my opinion, the research focuses on a topic of relevance, intending to advance on a condition with different gaps.
The limitations of this study are well described in the discussion, and the question mark in the title shows the existing incertitudes and invites to future research.
Author Response
Dear Reviewer,
Thank you very much for your time to review our manuscript entitled ‘’Can presepsin be of value in reducing unnecessary antibiotic exposure after birth?’’
Reviewer 2 Report
The manuscript in exam is a prospective cohort study. The authors evaluated the diagnostic accuracy of presepsin for culture-proven and culture-negative EOS in a cohort of infants with an indication for empirical antibiotics. According to the study results, the authors show that presepsin is a biomarker with acceptable diagnostics accuracy for EOS in preterm infants and can be of value in reducing antibiotic exposure after birth. In term born infants the diagnostic accuracy was low. In this study, a total of 65 term born infants and 15 preterm born infants were classified as EOS cases. So, considering that infants often start unnecessarily with antibiotics directly after birth due to the lack of rapid and accurate diagnostic tool for early-onset neonatal sepsis at initial suspicion, this study is interesting and useful. Then, the study sample is considerable for its large dimension. However, the text is not fluent and not free from grammatical errors in different points. For this reason, the submission to editing service is recommended because the manuscript, in this form, is not suitable for publication.
Author Response
Dear Reviewer,
Thank you very much for your time to review our manuscript entitled ‘’Can presepsin be of value in reducing unnecessary antibiotic exposure after birth?’’ We use the language editing service provided by MDPI to improve the text and grammatical errors.
Reviewer 3 Report
Comments
This article aimed to determine the diagnostic accuracy of presepsin for EOS before antibiotic initiation and investigate whether presepsin can be used to guide clinicians whether or not to start antibiotics. The author has done detailed work, but the study design must be major revised, see the following comments for details.
Study Design
1. Please provide the registration number for this multicenter prospective study
2. The flow chart of this study including inclusion criteria and exclusion criteria must be displayed, please refer to this article: Pospisilova I et al. Evaluation of presepsin as a diagnostic tool in newborns with risk of early-onset neonatal sepsis. Front Pediatr. 2023 Jan 9;10:1019825. doi: 10.3389/fped.2022.1019825.
3. What are the criteria for EOS suspicion? According to your method, I think your criteria is the Dutch EOS guideline. Please clarify it in the introduction and abstract.
4. The authors should compare the diagnostic value of presepsin to some biomarkers used in daily care, such as C-reactive protein, procalcitonin and interleukins.
5. Stratified analysis of gestational age: The primary cause of EOS is different in preterm and term infants, and I believe the two groups should be analyzed separately.
6. What is the test type? CLEIA or ELISA
Results
1. Please separate the analysis of preterm and term infants. For example, Table 1 and Figure 2 should also be presented separately for preterm and term infants.
2. Figure 2 should be graphed to show whether there is a significant difference between the two groups at different time points. Also, Figure 2 should precede Figure 1.
3. A table should be added to the text to illustrate the AUC, positive predictive value, negative predictive value and other indicators of presepsin measured at different time points to predict the diagnosis of EOS.
Discussion
1. What is the innovation of this study compared with other similar studies? Please highlight it in the first paragraph of this section.
2. The possible mechanisms of presepsin as a predictor of EOS should be discussed.
3. The structure of this part is confusing. It is suggested to discuss the similarities and differences between this article and other similar studies in the order of the diagnostic accuracy of term infants and the diagnostic accuracy of preterm infants.
Author Response
Reviewer 3
Comments
This article aimed to determine the diagnostic accuracy of presepsin for EOS before antibiotic initiation and investigate whether presepsin can be used to guide clinicians whether or not to start antibiotics. The author has done detailed work, but the study design must be major revised, see the following comments for details.
Dear Reviewer,
Thank you very much for your time to review our manuscript entitled ‘’Can presepsin be of value in reducing unnecessary antibiotic exposure after birth?” Each of your insights have served to strengthen our manuscript and we have made changes to reflect them. Below we present the detailed replies to your comments. Please find also the revised manuscript.
Study Design
- Please provide the registration number for this multicenter prospective study
Response of the authors: As this was an observational study, the study was not registered.
- The flow chart of this study including inclusion criteria and exclusion criteria must be displayed, please refer to this article: Pospisilova I et al. Evaluation of presepsin as a diagnostic tool in newborns with risk of early-onset neonatal sepsis. Front Pediatr. 2023 Jan 9;10:1019825. doi: 10.3389/fped.2022.1019825.
Response of the authors: A flow chart with study inclusions was added (Supplemental Figure 1). Besides, the study mentioned by the Reviewer is being discussed in the revised manuscript. Thank you for this suggestion.
- What are the criteria for EOS suspicion? According to your method, I think your criteria is the Dutch EOS guideline. Please clarify it in the introduction and abstract.
Response of the authors: The inclusion criterion was initiation of antibiotics for a suspicion of early-onset sepsis based on the Dutch EOS guideline. We added two supplemental Tables (Supplemental Table 6 and 7) with maternal risk factors and neonatal signs and symptoms mentioned in the guideline, clarifying when antibiotics are initiated according to this guideline.
- The authors should compare the diagnostic value of presepsin to some biomarkers used in daily care, such as C-reactive protein, procalcitonin and interleukins.
Response of the authors: Due to the limited amount of extra blood that we were allowed to collect, we were unable to perform other laboratory tests such as C-reactive protein, procalcitonin and different interleukins unfortunately. This is being discussed in the discussion: ‘’Furthermore, we did not compare the diagnostic accuracy of presepsin with other biomarkers such as CRP, PCT and IL-6, but other studies and meta-analysis, however, demonstrated a higher accuracy of presepsin compared to these other biomarkers at initial EOS suspicion.[32, 33]’’ (line 259-262).
- Stratified analysis of gestational age: The primary cause of EOS is different in preterm and term infants, and I believe the two groups should be analyzed separately.
Response of the authors: We completely agree with the reviewer that the primary cause of EOS differs between term and preterm born infants. For that reason analyses were performed separately for these two groups. The baseline results are demonstrated for these two groups separately as well in the revised version of the manuscript.
- What is the test type? CLEIA or ELISA
Response of the authors: The Pathfast is a chemiluminescent immunoassay analyser (CLEIA). This is clarified in the revised version of the manuscript (line 306).
Results
- Please separate the analysis of preterm and term infants. For example, Table 1 and Figure 2 should also be presented separately for preterm and term infants.
Response of the authors: results for figures are already separated for term and preterm born infants. In the revised manuscript table 1 is also presented separately for term and preterm infants.
- Figure 2 should be graphed to show whether there is a significant difference between the two groups at different time points. Also, Figure 2 should precede Figure 1.
Response of the authors: Presepsin concentrations were significantly higher at all time-points in both term and preterm born infants, except in umbilical cord blood in term born infants. This is demonstrated in the revised Figure. Besides, the order of the figures was adjusted according to the suggestion of the Reviewer.
- A table should be added to the text to illustrate the AUC, positive predictive value, negative predictive value and other indicators of presepsin measured at different time points to predict the diagnosis of EOS.
Response of the authors: A table demonstrating the AUC at all time-points was already presented. A table demonstrating the positive predictive value, negative predictive value, sensitivity and specificity at the Youden’s index are added for term (Supplemental Table 2) and preterm born infants (Supplemental Table 4) respectively.
Discussion
- What is the innovation of this study compared with other similar studies? Please highlight it in the first paragraph of this section.
Response of the authors: Previous studies are biased by the study design including healthy control infants without suspicion for EOS and/or excluding infants with clinical sepsis requiring antibiotics. This is the first study recruiting all infants starting with antibiotics for a suspicion of EOS. This approach minimizes bias and enables us to generalize results and applicability to clinical practise. This is highlighted in the first paragraph of the discussion in the revised manuscript (line 156-157).
- The possible mechanisms of presepsin as a predictor of EOS should be discussed.
Response of the authors: Possible mechanisms of action of presepsin are explained in the introduction: After binding of bacterial ligands to the cell surface of monocytes and macrophages, CD14 is shedding from the cell surface and is subject to proteolysis.[11, 12] This leads to release of various fragments and finally generation of a small soluble peptide structure (64 amino acids, 13kDa) named soluble CD14 subtype (sCD14-ST) or presepsin.[13] (line 65-68).
In the revised manuscript potential advantages of presepsin as a predictor of EOS are discussed: Presepsin may be a rapid and early biomarker for EOS. Presepsin concentrations can be measured quickly as point-of-care-test (POCT), a limited blood volume is needed and concentrations increase early and rapidly in case of a bacterial infection (line 195-198).
- The structure of this part is confusing. It is suggested to discuss the similarities and differences between this article and other similar studies in the order of the diagnostic accuracy of term infants and the diagnostic accuracy of preterm infants.
Response of the authors: In the revised discussion of the manuscript, the studies on presepsin for EOS in term born infants are being discussed first. This is followed by a discussion of the papers investigating the diagnostic accuracy for EOS in preterm born infants (line167-180)
Round 2
Reviewer 3 Report
It is my honor to review this manuscript. The authors' revised manuscript is much improved and it is suitable for publication.